# OGB-LSC: A Large-Scale Challenge for Machine Learning on Graphs

**Weihua Hu[1], Matthias Fey[2], Hongyu Ren[1], Maho Nakata[3], Yuxiao Dong[4], Jure Leskovec[1]**
[1]Department of Computer Science, Stanford University
[2]Department of Computer Science, TU Dortmund University
[3]RIKEN, [4]Facebook AI
ogb-lsc@cs.stanford.edu

## Abstract

Enabling effective and efficient machine learning (ML) over large-scale graph data (*e.g.*, graphs with billions of edges) can have a great impact on both industrial and scientific applications. However, existing efforts to advance large-scale graph ML have been largely limited by the lack of a suitable public benchmark. Here we present OGB Large-Scale Challenge (OGB-LSC), a collection of three real-world datasets for facilitating the advancements in large-scale graph ML. The OGB-LSC datasets are orders of magnitude larger than existing ones, covering three core graph learning tasks—link prediction, graph regression, and node classification. Furthermore, we provide dedicated baseline experiments, scaling up expressive graph ML models to the massive datasets. We show that expressive models significantly outperform simple scalable baselines, indicating an opportunity for dedicated efforts to further improve graph ML at scale. Moreover, OGB-LSC datasets were deployed at ACM KDD Cup 2021 and attracted more than 500 team registrations globally, during which significant performance improvements were made by a variety of innovative techniques. We summarize the common techniques used by the winning solutions and highlight the current best practices in large-scale graph ML. Finally, we describe how we have updated the datasets after the KDD Cup to further facilitate research advances. The OGB-LSC datasets, baseline code, and all the information about the KDD Cup are available at https://ogb.stanford.edu/docs/lsc/.

## 1 Introduction

Machine Learning (ML) on graphs has attracted immense attention in recent years because of the prevalence of graph-structured data in real-world applications. Modern application domains include Web-scale social networks (Ugander *et al.*, 2011), recommender systems (Ying *et al.*, 2018), hyperlinked Web documents (Kleinberg, 1999), knowledge graphs (KGs) (Bollacker *et al.*, 2008; Vrandečić and Krötzsch, 2014), as well as the molecule simulation data generated by the ever-increasing scientific computation (Nakata and Shimazaki,

Table 1: **Basic statistics of the OGB-LSC datasets used in KDD Cup 2021.** Datasets marked by † has been updated to v2 after the KDD Cup (*cf.* Section 3).

| Task type | Dataset | Statistics | |
|---|---|---|---|
| Node-level | MAG240M | #nodes: | 244,160,499 |
| | | #edges: | 1,728,364,232 |
| Link-level | WikiKG90M† | #nodes: | 87,143,637 |
| | | #edges: | 504,220,369 |
| Graph-level | PCQM4M† | #graphs: | 3,803,453 |
| | | #edges (total): | 55,399,880 |

2017; Chanussot *et al.*, 2021). All these domains involve large-scale graphs with billions of edges or a dataset with millions of graphs. Deploying accurate graph ML at scale will have a huge practical im-

35th Conference on Neural Information Processing Systems (NeurIPS 2021) Track on Datasets and Benchmarks.

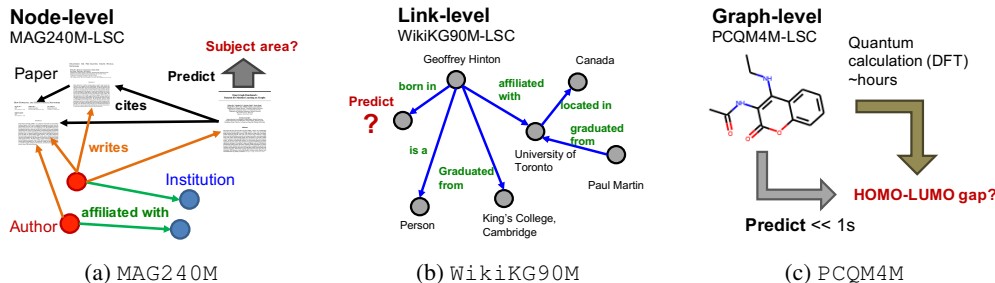

(a) MAG240M       (b) WikiKG90M       (c) PCQM4M

Figure 1: **Overview of the three OGB-LSC datasets, covering node-, link-, and graph-level prediction tasks, respectively. (a)** MAG240M is a heterogeneous academic graph, and the task is to predict the subject areas of papers situated in the heterogeneous graph (*cf.* Section 2.1). **(b)** WikiKG90M is a knowledge graph, and the task is to impute missing triplets (*cf.* Section 2.2). **(c)** PCQM4M is a quantum chemistry dataset, and the task is to predict an important molecular property—the HOMO-LUMO gap—of a given molecule (*cf.* Section 2.3).

pact, enabling better recommendation results, improved web document search, more comprehensive KGs, and accurate ML-based drug and material discovery.

However, community efforts to advance state-of-the-art in large-scale graph ML have been quite limited. In fact, most of graph ML models have been developed and evaluated on extremely small datasets (Yang *et al.*, 2016; Morris *et al.*, 2020; Bordes *et al.*, 2013). Recently, the Open Graph Benchmark (OGB) has been introduced to provide a collection of larger graph datasets (Hu *et al.*, 2020a), but they are still small compared to graphs found in the industrial and scientific applications.

Handling large-scale graphs is challenging, especially for state-of-the-art expressive Graph Neural Networks (GNNs) (Kipf and Welling, 2017; Hamilton *et al.*, 2017; Velickovic *et al.*, 2018) because they make predictions on each node based on the information from many other nodes. Effectively training these models at scale requires sophisticated algorithms that are well beyond standard SGD over i.i.d. data (Hamilton *et al.*, 2017; Chen *et al.*, 2018; Chiang *et al.*, 2019; Zeng *et al.*, 2020). More recently, researchers improve the model scalability by significantly simplifying GNNs (Wu *et al.*, 2019; Rossi *et al.*, 2020; Huang *et al.*, 2020), which inevitably limits their expressive power.

However, in deep learning, it has been demonstrated over and over again that one needs big expressive models and train them on big data to achieve the best performance (He *et al.*, 2016; Russakovsky *et al.*, 2015; Vaswani *et al.*, 2017; Devlin *et al.*, 2018; Brown *et al.*, 2020). In graph ML, the trend has been the opposite—models get simplified and less expressive to be able to scale to large graphs (Wu *et al.*, 2019). Thus, there is a massive opportunity to enable graph ML techniques to work with realistic and large-scale graph datasets, exploring the potential of expressive models for big graphs.

Here we present a large-scale graph ML challenge, OGB Large-Scale Challenge (OGB-LSC), to facilitate the development of state-of-the-art graph ML models for massive modern datasets. Specifically, we introduce three large-scale, realistic, and challenging datasets—MAG240M, WikiKG90M, and PCQM4M—that are unprecedentedly large in scale (see Table 1; the sizes are 10 to 100 times larger than the corresponding original OGB datasets[1]) and cover predictions at the level of nodes, links, and graphs, respectively. An overview of the datasets is provided in Figure 1.

Beyond providing the datasets, we perform an extensive baseline analysis on each dataset and implement both simple baseline models and advanced expressive models at scale. We find that advanced expressive models—despite requiring more efforts to scale up—do benefit from the large data and significantly outperform simple baseline models that are easy to scale.

To facilitate the community engagement, we recently organized the ACM KDD Cup 2021 around the OGB-LSC datasets. The competition attracted more than 500 team registrations and 150 leaderboard submissions. Within the three-month duration of the competition (March 15 to June 15, 2021), we have already witnessed innovative methods being developed to provide impressive performance

---

[1]Specifically, MAG240M is 126 times larger than ogbn-mag in terms of the number of nodes, WikiKG90M is 35 times larger than ogbl-wikikg2 in terms of the number of nodes, and PCQM4M is 9 times larger than ogbg-molpcba in terms of the number of graphs.

gains[2], further solidifying the value of the OGB-LSC datasets to advance state-of-the-art. We summarize the common techniques shared by the winning solutions, highlighting the current best practices of large-scale graph ML. Moreover, based on the lessons learned from the KDD Cup, we describe the future plan to update the datasets so that they can be further used to advance large-scale graph ML.

## 2 OGB-LSC Datasets, Baselines, and KDD Cup Summary

We describe the OGB-LSC datasets, covering three key task categories (node-, link-, and graph-level prediction tasks) of ML on graphs. We emphasize the practical relevance and data split for each dataset, making our task closely aligned to realistic applications. Through our extensive baseline experiments, we show that advanced expressive models tend to give much better performance than simple graph ML models, leaving room for further improvement. All the OGB-LSC datasets are available through the OGB Python package (Hu *et al.*, 2020a). All the baseline and package code is available at `https://github.com/snap-stanford/ogb`.

In addition, we highlight the top 3 winning results from our KDD Cup 2021 that significantly advance state-of-the-art and summarize common techniques used by the winning solutions. Note that while our baselines only used a single model for simplicity, all the winners used extensive model ensembling for their test submissions in order to maximize the performance. For a more direct comparison, we also report the winners' self-reported validation accuracy in the main text, which still exhibits significant accuracy improvement over our strong baselines.

### 2.1 `MAG240M`: Node-Level Prediction

**Practical relevance and dataset overview**. The volume of scientific publication has been increasing exponentially, doubling every 12 years (Dong *et al.*, 2017). Currently, subject areas of ARXIV papers are manually determined by the paper's authors and ARXIV moderators. An accurate automatic predictor of papers' subject categories not only reduces the significant burden of manual labeling, but can also be used to classify the vast number of non-ARXIV papers, thereby allowing better search and organization of academic papers.

MAG240M is a heterogeneous academic graph extracted from the Microsoft Academic Graph (MAG) (Wang *et al.*, 2020). Given arXiv papers situated in the heterogeneous graph, whose schema diagram is illustrated in Figure 2, we aim to automatically annotate their topics, *i.e.*, predicting the primary subject area of each ARXIV paper.

**Graph**. We extract 121M academic papers in English from MAG (version: 2020-11-23) to construct a heterogeneous academic graph. The resultant paper set is written by 122M author entities, who are affiliated with 26K institutes. Among these papers, there are 1.3 billion citation links captured by MAG. Each paper is associated with its natural language title and most papers' abstracts are also available. We concatenate the title and abstract by period and pass it to a ROBERTA sentence encoder (Liu *et al.*, 2019; Reimers and Gurevych, 2019), generating a 768-dimensional vector for each paper node. Among the 121M paper nodes, approximately 1.4M nodes are ARXIV papers annotated with 153 ARXIV subject areas, *e.g.*, cs.LG (Machine Learning). On the paper nodes, we attach the publication years as meta information.

**Prediction task and evaluation metric**. The task is to predict the primary subject areas of the given ARXIV papers, which is cast as an ordinary multi-class classification problem. The metric is the classification accuracy.

To understand the relation between the prediction task and the heterogeneous graph structure, we analyze the graph homophily (McPherson *et al.*, 2001)—tendency of two adjacent nodes to share the same labels—to better understand the interplay between heterogeneous graph connectivity and the prediction task. Homophily is normally analyzed over a homogeneous graph, but we extend the analysis to the heterogenous graph by considering meta-paths (Sun *et al.*, 2011)—a path consisting of a sequence of relations defined between different node types. Given a meta-path, we can say two nodes are adjacent if they are connected by the meta-path. Table 3 shows the homophily for different kinds of meta-paths with different levels of connection strength. Compared to the direct

---

[2]See the results at `https://ogb.stanford.edu/kddcup2021/results/`

citation connection (*i.e.*, P-P), certain meta-paths (*i.e.*, P-A-P) give rise to much higher degrees of homophiliness, while other meta-paths (*i.e.*, P-A-I-A-P) provide much less homophily. As homophily is the central graph property exploited by many graph ML models, we believe that discovering essential heterogeneous connectivity is important to achieve good performance on this dataset.

**Dataset split**. We split the data according to time. Specifically, we train models on ARXIV papers published until 2018, validate the performance on the 2019 papers, and finally test the performance on the 2020 papers. The split reflects the practical scenario of helping the authors and moderators annotate the subject areas of the newly-published ARXIV papers.

**Baseline**. We benchmark a broad range of graph ML models in both homogeneous (where only paper to paper relations are considered) and full heterogeneous settings. For both settings, we convert the directed graph into an undirected graph for simplicity. First, for the homogeneous setting, we benchmark the simple baseline models: graph-agnostic MLP, Label Propagation, and the recently-proposed simplified graph methods: SGC (Wu *et al.*, 2019), SIGN (Rossi *et al.*, 2020) and MLP+C&S (Huang *et al.*, 2020), which are inherently scalable by decoupling predictions from propagation. Furthermore, we benchmark state-of-the-art expressive GNNs trained with neighborhood sampling (NS) (Hamilton *et al.*, 2017), where we recursively sample 25 neighbors in the first layer and 15 neighbors in the second layer during training time. At inference time, we sample at most 160 neighbors for each layer. Here, we benchmark two types of strong models: the GRAPHSAGE (Hamilton *et al.*, 2017) model (performing mean aggregation and utilizing skip-connections), and the more advanced GRAPH ATTENTION NETWORK (GAT) model (Velickovic *et al.*, 2018). For the full heterogeneous setting, we follow Schlichtkrull *et al.* (2018) and learn distinct weights for each individual relation type (denoted by R-GRAPHSAGE and R-GAT, where "R" stands for "Relational"). We obtain the input features of authors and institutions by averaging the features of papers belonging to the same author and institution, respectively. The models are trained with NS. We note that the expressive GNNs trained with NS require more efforts to scale up, but are more expressive than the simple baselines.

**Hyper-parameters**. Hyper-parameters are selected based on their best validation performance. For all the models without NS, we tuned the hidden dimensionality $\in \{128, 256, 512, 1024\}$, MLP depth $\in \{1, 2, 3, 4\}$, dropout ratio $\in \{0, 0.25, 0.5\}$, propagation layers (for SGC, SIGN, and C&S) $\in \{2, 3\}$. For all the GNN models with NS, we use a hidden dimensionality of 1024. We make use of batch normalization (Ioffe and Szegedy, 2015) and ReLU activation in all models.

**Discussion**. Validation and test performances of all models considered are shown in Table 2. First, the graph-agnostic MLP and Label Propagation algorithm perform poorly, indicating that both graph structure and feature information are indeed important for the given task. Across the graph ML models operating on the homogeneous paper graph, GNNs with NS perform the best, with slight gains compared to their simplified versions. In particular, the advanced expressive graph attention aggregation is favourable compared to the uniform mean aggregation in GRAPHSAGE. Furthermore, considering all available heterogeneous relational structure in the heterogeneous graph setting yields significant improvements, with performance gains up to 3 percentage points. Again, the advanced attention aggregation provides favorable performance. Overall, our experiments highlight the benefits of developing and evaluating advanced expressive models on the larger scale.

**KDD Cup 2021 summary**. In Table 2, we show the results of the top 3 winners of the KDD Cup: BD-PGL Team (Shi *et al.*, 2021), Academic Team (Addanki *et al.*, 2021), and Synerise AI Team (Daniluk *et al.*, 2021). All the solutions outperform our baselines significantly, yielding 5–6% gain in test accuracy. For a more direct comparison, with a single model (no model ensembling), the BD-PGL Team reports a validation accuracy of 73.71% (Shi *et al.*, 2021), improving our best R-GAT baseline by 3.7%.

Notably, all the winning solutions used the target labels as input to their models, which allows the models to propagate labels together with the features. Regarding the GNN architectures, the BD-PGL adopted the expressive Transformer-based UniMP architecture (Shi *et al.*, 2020), while the Academic adopted the standard MPNN (Gilmer *et al.*, 2017) but trained it with self-supervised contrastive learning on unlabeled paper nodes (Thakoor *et al.*, 2021). These results suggest that expressive GNNs are indeed promising for this dataset. Finally, both the BD-PGL and Academic teams exploited the temporal aspect of the academic graph by using the publication years either as input positional encoding (Shi *et al.*, 2021) or as a way to sample mini-batch subgraphs for

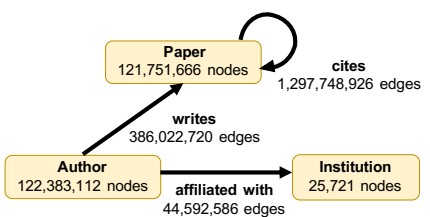

Figure 2: **A schema diagram of MAG240M.**

Table 3: **Analysis of graph homophily for different meta-paths connecting 1,251,341 arXiv papers (only train+validation).** Connection strength indicates the number of different possible paths along the template meta-path, *e.g.*, meta-path "Paper-Author-Paper (P-A-P)" with connection strength 3 means that at least 3 authors are shared for the two papers of interest. Homophily ratio is the ratio of two nodes having the same target labels.

Table 2: **Results of MAG240M measured by the accuracy (%).**

| Model | #Params | Validation | Test |
|---|---|---|---|
| MLP | 0.5M | 52.67 | 52.73 |
| LABELPROP | 0 | 58.44 | 56.29 |
| SGC | 0.7M | 65.82 | 65.29 |
| SIGN | 3.8M | 66.64 | 66.09 |
| MLP+C&S | 0.5M | 66.98 | 66.18 |
| GRAPHSAGE (NS) | 4.9M | 66.79 | 66.28 |
| GAT (NS) | 4.9M | 67.15 | 66.80 |
| R-GRAPHSAGE (NS) | 12.2M | 69.86 | 68.94 |
| R-GAT (NS) | 12.3M | **70.02** | **69.42** |
| KDD 1ST: BD-PGL | | | **75.49** |
| KDD 2ND: ACADEMIC | | | **75.19** |
| KDD 3RD: SYNERISE AI | | | **74.60** |

| Meta-path | Connect. strength | Homophily ratio (%) | #Edges |
|---|---|---|---|
| P-P | 1 | 57.80 | 2,017,844 |
| P-A-P | 1 | 46.12 | 88,099,071 |
| | 2 | 57.02 | 12,557,765 |
| | 4 | 64.03 | 1,970,761 |
| | 8 | 66.65 | 476,792 |
| | 16 | 70.46 | 189,493 |
| P-A-I-A-P | 1 | 3.83 | 159,884,165,669 |
| | 2 | 4.61 | 81,949,449,717 |
| | 4 | 5.69 | 33,764,809,381 |
| | 8 | 6.85 | 12,390,929,118 |
| | 16 | 7.70 | 4,471,932,097 |
| All pairs | 0 | 1.99 | 782,926,523,470 |

GNNs (Addanki *et al.*, 2021). As real-world large-scale graphs are almost always dynamic, exploiting the temporal information is a promising direction of future research.

## 2.2 WikiKG90M: Link-Level Prediction

**Practical relevance and dataset overview**. Large encyclopedic Knowledge Graphs (KGs), such as Wikidata (Vrandečić and Krötzsch, 2014) and Freebase (Bollacker *et al.*, 2008), represent factual knowledge about the world through triplets connecting different entities, *e.g.*, *Hinton* $\xrightarrow{citizen\ of}$ *Canada*. They provide rich structured information about many entities, aiding a variety of knowledge-intensive downstream applications such as information retrieval, question answering (Singhal, 2012), and recommender systems (Guo *et al.*, 2020). However, these large KGs are known to be far from complete (Min *et al.*, 2013), missing many relational information between entities.

WikiKG90M is a Knowledge Graph (KG) extracted from the *entire* Wikidata knowledge base. The task is to automatically impute missing triplets that are not yet present in the current KG. Accurate imputation models can be readily deployed on the Wikidata to improve its coverage.

**Graph**. Each triplet (head, relation, tail) in WikiKG90M represents an Wikidata claim, where head and tail are the Wikidata items, and relation is the Wikidata predicate. We extracted triplets from the public Wikidata dump downloaded at three time-stamps: September 28, October 26, and November 23 of 2020, for training, validation, and testing, respectively. We retain all the entities and relations in the September dump, resulting in 87,143,637 entities, 1,315 relations, and 504,220,369 triplets in total.

In addition to extracting triplets, we provide text features for entities and relations. Specifically, each entity/relation in Wikidata is associated with a title and a short description, *e.g.*, one entity is associated with the title 'Geoffrey Hinton' and the description 'computer scientist and psychologist'. Similar to MAG240M, we provide ROBERTA embeddings (Reimers and Gurevych, 2019; Liu *et al.*, 2019) as node and edge features.[3]

---

[3]We concatenate the title and description with comma, *e.g.*, 'Geoffrey Hinton, computer scientist and psychologist', and pass the sentence to a ROBERTA sentence encoder (Note that the ROBERTA model was trained before September 2020, so there is no obvious information leak). The title or/and description are sometimes missing, in which case we simply use the blank sentence to replace it.

**Prediction task and evaluation metric**. The task is the KG completion, *i.e.*, given a set of training triplets, predict a set of test triplets. For evaluation, we follow the protocol similar to how KG completion is evaluated (Bordes *et al.*, 2013). Specifically, for each validation/test triplet, (head, relation, tail), we corrupt tail with randomly-sampled 1000 negative entities, *e.g.*, tail_neg, such that (head, relation, tail_neg) does not appear in the train/validation/test KG. The model is asked to rank the 1001 candidates (consisting of 1 positive and 1000 negatives) for each triplet and predict the top 10 entities that are most likely to be positive. The goal is to rank the ground-truth positive entity as high in the rank as possible, which is measured by Mean Reciprocal Rank (MRR). [4]

**Dataset split**. We split the triplets according to time, simulating a realistic KG completion scenario of imputing missing triplets not present at a certain timestamp. Specifically, we construct three KGs using the aforementioned September, October, and November KGs, where we only retain entities and relation types that appear in the earliest September KG. We use the triplets in the September KG for training, and use the additional triplets in the October and November KGs for validation and test, respectively.

We analyze the effect of the time split. We find that head entities of validation triplets tend to be less popular entities; on average, they only have 6.5 out-degrees in the training KG, which is less than a quarter of the out-degree averaged over training triplets (*i.e.*, 28.0). This suggests that learning signals for predicting validation (and test) triplets are sparse. Nonetheless, even for the sparsely-connected triplets, we find the textual information provides important clues, as illustrated in Table 4. Hence, we expect that advanced graph models that effectively incorporate textual information will be key to achieve good performance on the challenging time split.

**Baseline**. We consider two representative KG embedding models: TRANSE (Bordes *et al.*, 2013) and COMPLEX (Trouillon *et al.*, 2016). These models define their own *decoders* to score knowledge triplets using the corresponding entity and relation embeddings. For instance, TRANSE uses $-\|\boldsymbol{h} + \boldsymbol{r} - \boldsymbol{t}\|_2$ as the decoder, where $\boldsymbol{h}$, $\boldsymbol{r}$, and $\boldsymbol{t}$ are embeddings of head, relation, and tail, respectively. For the encoder function (mapping each entity and relation to its embedding), we consider the following three options. **Shallow:** We use the distinct embedding for each entity and relation, as normally done in KG embedding models. **RoBERTa:** We use two MLP encoders (one for entity and another for relation) that transform the ROBERTA features into entity and relation embeddings. **Concat:** To enhance the expressive power of the previous encoder, we concatenate the shallow learnable embeddings into the ROBERTA features, and use the MLPs to transform the concatenated vectors to get the final embeddings. This way, the MLP encoders can adaptively utilize the ROBERTA features and the shallow embeddings to fit the large amount of triplet data. To implement our baselines, we utilize DGL-KE (Zheng *et al.*, 2020).

**Hyper-parameters**. For the loss function, we use the negative sampling loss from Sun *et al.* (2019), where we pick margin $\gamma$ from $\{1,4,8,10,100\}$. In order to balance the performance and the memory cost, we use the embedding dimensionality of 200 for all the models.

**Discussion**. Table 5 shows the validation and test performance of the six different models, *i.e.*, combination of two decoders (TRANSE and COMPLEX) and three encoders (SHALLOW, ROBERTA, and CONCAT). Notably, in terms of the encoders, we see that the most expressive CONCAT outperforms both SHALLOW and ROBERTA, indicating that both the textual information (captured by the ROBERTA embeddings) and structural information (captured by node-wise learnable embeddings) are useful in predicting validation and test triplets. In terms of the decoders, TRANSE and COMPLEX show similar performance with the CONCAT encoder, while they show somewhat mixed results with the SHALLOW and ROBERTA encoders.

Overall, our experiments suggest that the expressive encoder that combines both textual information and structural information gives the most promising performance. In the KG completion literature, the design of the encoder has been much less studied compared to the decoder designs. Therefore, we believe there is a huge opportunity in scaling up more advanced encoders, especially GNNs (Schlichtkrull *et al.*, 2018), to further improve the performance on this dataset.

**KDD Cup 2021 summary**. Table 5 shows the results of the top 3 winners of the KDD Cup: BD-PGL Team (Su *et al.*, 2021), OhMyGod Team (Peng *et al.*, 2021), and the GraphMIRAcles Team (Cai *et al.*, 2021). All the winning solutions outperform our strong baselines significantly,

---

[4]Note that this is more strict than the standard MRR since there is no partial score for positive entities being ranked outside of top 10.

Table 4: **Textual representation of validation triplets whose head entities only *appear once* as head in the training `WikiKG90M`.**

| Head | Relation | Tail |
|------|----------|------|
| Food and drink companies of Bulgaria | combines topics | Bulgaria |
| Performing arts in Denmark | combines topics | performing arts |
| Anglicanism in Grenada | combines topics | Anglicanism |
| Chuan Li | occupation | researcher |
| Petra Junkova | given name | Petra |

Table 5: **Results of `WikiKG90M` measured by Mean Reciprocal Rank (MRR).**

| Model | #Params | Validation | Test |
|-------|---------|------------|------|
| TRANSE-SHALLOW | 17.4B | 0.7559 | 0.7412 |
| COMPLEX-SHALLOW | 17.4B | 0.6142 | 0.5883 |
| TRANSE-ROBERTA | 0.3M | 0.6039 | 0.6288 |
| COMPLEX-ROBERTA | 0.3M | 0.7052 | 0.7186 |
| TRANSE-CONCAT | 17.4B | **0.8494** | 0.8548 |
| COMPLEX-CONCAT | 17.4B | 0.8425 | **0.8637** |
| KDD 1ST: BD-PGL | | | **0.9727** |
| KDD 2ND: OHMYGOD | | | **0.9712** |
| KDD 3RD: GRAPHMIRACLES | | | **0.9707** |

achieving near-perfect test MRR score of 0.97. For a more direct comparison, with a single model (no model ensembling), the BD-PGL Team reports a validation MRR of 0.92 (Su *et al.*, 2021), improving our best COMPLEX-CONCAT baseline by 0.07 points in validation MRR. Similar to our baselines, all the winners utilize the KG embedding approach as the backbone, and adopt the encoder that takes both shallow embedding and textual embeddings into account. Specifically, BD-PGL proposed the NOTE model (Su *et al.*, 2021) which makes the ROTATE model more expressive, while OhMyGod adopted the ensemble of several existing KG embedding models. On the other hand, GraphMIRAcles explored different design choices for the encoder and found that adding residual connection for shallow embeddings significantly improved the model performance.

In addition to the model advances, all the winners exploited some statistical property of candidate tail entities. Most notably, Yang *et al.* (2021) found that simply by sorting the candidate tails by the frequency they appear in the training KG, it was possible to achieve validation MRR of 0.75, rivaling our TRANSE-SHALLOW baseline. This highlights that our negative tail candidates are mostly rare entities that can be easily distinguished from the true tail entity. On the other hand, the practical KG completion presents a much harder challenge: the candidate tails are *not* provided, and a model needs to predict the true tail entity *out of all the possible 87M entities.* As the performance on `WikiKG90M` has already saturated, we have updated `WikiKG90M` to `WikiKG90Mv2` to reflect the realistic setting in large-scale KG completion. See Section 3 for further details.

### 2.3 `PCQM4M`: Graph-Level Prediction

**Practical relevance and dataset overview**. Density Functional Theory (DFT) is a powerful and widely-used quantum physics calculation that can accurately predict various molecular properties such as the shape of molecules, reactivity, responses by electromagnetic fields (Burke, 2012). However, DFT is time-consuming and takes up to several hours per small molecule. Using fast and accurate ML models to approximate DFT enables diverse downstream applications, such as property prediction for organic photovaltaic devices (Cao and Xue, 2014) and structure-based virtual screening for drug discovery (Ferreira *et al.*, 2015).

`PCQM4M` is a quantum chemistry dataset originally curated under the PubChemQC project (Nakata, 2015; Nakata and Shimazaki, 2017). Based on the PubChemQC, we define a meaningful ML task of predicting DFT-calculated HOMO-LUMO energy gap of molecules given their 2D molecular graphs. The HOMO-LUMO gap is one of the most practically-relevant quantum chemical properties of molecules since it is related to reactivity, photoexcitation, and charge transport (Griffith and Orgel, 1957). Moreover, predicting the quantum chemical property only from 2D molecular graphs without their 3D equilibrium structures is also practically favorable. This is because obtaining 3D equilibrium structures requires DFT-based geometry optimization, which is expensive on its own.

To ensure the resulting models are practically useful, we limit the average inference budget per molecule (including both pre-processing and model inference) to be less than 0.1 second using a single GPU and CPU (multi-threading on a multi-core CPU is allowed). This means that expensive (quantum) calculations cannot be used to perform inference. As our test set contains 377,423 molecules, we require the all the prediction to be made within 12 hours. Note that this time constraint is quite generous for ordinary GNNs—each of our baseline GNN only took about 3 minutes to perform inference over the entire test data.

**Graph**. We provide molecules as the SMILES strings (Weininger, 1988), from which 2D molecule graphs (nodes are atoms and edges are chemical bonds) as well as molecular fingerprints (hand-engineered molecular feature developed by the chemistry community) can be obtained. By default, we follow OGB (Hu *et al.*, 2020a) to convert the SMILES string into a molecular graph representation, where each node is associated with a 9-dimensional feature (*e.g.*, atomic number, chirality) and each edge comes with a 3-dimensional feature (*e.g.*, bond type, bond stereochemistry), although the optimal set of input graph features remains to be explored.

**Prediction task and evaluation metric**. The task is graph regression: predicting the HOMO-LUMO energy gap in electronvolt (eV) given 2D molecular graphs. Mean Absolute Error (MAE) is used as evaluation metric.

**Dataset split**. We split molecules by their PubChem ID (CID) with ratio 80/10/10. Our original intention was to provide the scaffold split (Hu *et al.*, 2020a; Wu *et al.*, 2018), but the provided data turns out to be split by the CID due to some pre-processing bug. The CID number itself does not indicate particular meaning about the molecule, but splitting by CID may provide a moderate distribution shift (most likely not as severe as the scaffold split). We empirically compared the CID and scaffold splits and found the model performances were consistent between the two splits.[5]

**Baseline**. We benchmark two types of models: a simple MLP over the Morgan fingerprint (Morgan, 1965) and more advanced GNN models. For GNNs, we use the four strong models developed for graph-level prediction: Graph Convolutional Network (GCN) (Kipf and Welling, 2017) and Graph Isomorphism Network (GIN) (Xu *et al.*, 2019), as well as their variants, GCN-VIRTUAL and GIN-VIRTUAL, which augment graphs with a virtual node that is bidirectionally connected to all nodes in the original graph (Gilmer *et al.*, 2017). Adding the virtual node is shown to be effective across a wide range of graph-level prediction datasets in OGB (Hu *et al.*, 2020a). Edge features are incorporated following Hu *et al.* (2020b). At inference time, the model output is clamped between 0 and 50 to avoid model's anomalously large/small prediction.

**Hyper-parameters**. For the MLP over Morgan fingerprint, we set the fingerprint dimensionality to be 2048, and tune the fingerprint radius $\in \{2, 3\}$, as well as MLP's hyper-parameters: hidden dimensionality $\in \{1200, 1600\}$, number of hidden layers $\in \{2, 4, 6\}$, and dropout ratio $\in \{0, 0.2\}$. For GNNs, we tune hidden dimensionality, *i.e.*, width $\in \{300, 600\}$, number of GNN layers, *i.e.*, depth $\in \{3, 5\}$. Simple summation is used for graph-level pooling. For all MLPs (including GIN's), we use batch normalization (Ioffe and Szegedy, 2015) and ReLU activation.

**Discussion**. The validation and test results are shown in Table 6. We see both the GNN models significantly outperform the simple fingerprint baseline. Expressive GNNs (GIN and GIN-VIRTUAL) outperform less expressive ones (GCN and GCN-VIRTUAL); especially, the most advanced and expressive GIN-VIRTUAL model significantly outperforms the other GNNs. Nonetheless, the current performance is still much worse than the chemical accuracy of 0.043eV—an indicator of practical usefulness established by the chemistry community. In the same Table 6, we show our ablation, where we use only 10% of data to train the GIN-VIRTUAL model. We see the performance significantly deteriorate, indicating the importance of training the model on large data. Finally, in Table 7, we show the relation between model sizes and validation performance. We see that the largest models always achieve the best performance.

Overall, we find that advanced, expressive, and large GNN model gives the most promising performance on the PCQM4M dataset, although the performance still needs to be improved for practical use. We believe further advances in advanced modeling, expressive architectures, and larger model sizes could yield breakthrough in the large-scale molecular property prediction task.

**KDD Cup 2021 summary**. In Table 6, we show the results of the top 3 winners of the KDD Cup: Machine Learning Team (Ying *et al.*, 2021b), SuperHelix Team (Zhang *et al.*, 2021), and Quantum Team (Addanki *et al.*, 2021). The winners have significantly reduced the MAE compared our baselines, yielding around 0.03 points improvement in test MAE. For a more direct comparison, with a single model, the Machine Learning reports the validation MAE of 0.097 for their Graphormer model (Ying *et al.*, 2021a), which is 0.04 points lower than our best GIN-VIRTUAL baseline.

---

[5]Detailed discussion can be found at `https://github.com/snap-stanford/ogb/discussions/162`

Table 6: **Results of `PCQM4M` measured by MAE [eV].** The lower, the better. Ablation study of using only 10% of training data is also shown. Chemical accuracy indicates the final goal for practical usefulness.

| Model | #Params | Validation | Test |
|---|---|---|---|
| MLP-FINGERPRINT | 16.1M | 0.2044 | 0.2070 |
| GCN | 2.0M | 0.1684 | 0.1842 |
| GCN-VIRTUAL | 4.9M | 0.1510 | 0.1580 |
| GIN | 3.8M | 0.1536 | 0.1685 |
| GIN-VIRTUAL | 6.7M | **0.1396** | **0.1494** |
| MLP-FINGERPRINT (10% train) | 6.8M | 0.2708 | 0.2659 |
| GIN-VIRTUAL (10% train) | 6.7M | 0.1790 | 0.1892 |
| KDD 1ST: MACHINELEARNING | | | **0.1208** |
| KDD 2ND: SUPERHELIX | | | **0.1210** |
| KDD 3RD: QUANTUM | | | **0.1211** |
| Chemical accuracy (goal) | – | | 0.0430 |

Table 7: **Model size and the MAE performance [eV].** For both models, the width indicates the hidden dimensionality. For GIN-VIRTUAL, the depth represents the number of GNN layers, while for the MLP-FINGERPRINT, the depth represents the the number of hidden layers in MLP.

| Model | Width | Depth | #Params | Validation |
|---|---|---|---|---|
| MLP-FINGERPRINT | **1600** | **6** | 16.1M | **0.2044** |
| | **1600** | **4** | 11.0M | **0.2044** |
| | 1600 | 2 | 5.8M | 0.2220 |
| | 1200 | 6 | 9.7M | 0.2083 |
| GIN-VIRTUAL | **600** | **5** | 6.7M | **0.1410** |
| | 600 | 3 | 3.7M | 0.1462 |
| | 300 | 5 | 1.7M | 0.1442 |
| | 300 | 3 | 1.0M | 0.1512 |

In terms of methodology, we find that the winning solutions share three important components in common. (**1**) Their winning GNN models are indeed large and deep; the number of learnable parameters (single model) ranges from 50M up to 450M, while the number of GNN layers ranges from 11 up to 50, being significantly larger than our baseline models. (**2**) All the GNNs perform *global message passing* at each layer, either through the virtual nodes (Gilmer *et al.*, 2017) or fully-connected Transformer-style self-attention (Ying *et al.*, 2021a). (**3**) All the winners utilize 3D structure of molecules to supervise their GNNs. As 3D structure was not provided at our KDD Cup, the winners generate the 3D structure themselves using RDkit (Landrum *et al.*, 2006) or PySCF (Sun *et al.*, 2020), both of which provide cheap but less accurate 3D structure of molecules.

As modeling 3D molecular graphs is a promising direction in graph ML (Schütt *et al.*, 2017; Klicpera *et al.*, 2020; Sanchez-Gonzalez *et al.*, 2020; Hu *et al.*, 2021), we have updated PCQM4M to PCQM4Mv2 to include DFT-calculated 3D structures for training molecules. Details are provided in Section 3.

## 3 Updates after the KDD Cup

To facilitate further research advances, we have updated the datasets and leaderboards based on the lessons learned from our KDD Cup 2021. Here we briefly describe our updates. More details are provided in Appendix C.

**Updates on `WikiKG90M`.** From the KDD Cup results, we learned that most of our provided negative entities in the large-scale WikiKG90M are "easy negatives", and our current task gives overly-optimistic performance scores. In a realistic large-scale KG completion setting, ML models are required to predict the true tail entity from *nearly 90M entities*, which is much more challenging. To reflect this challenge, we have updated WikiKG90M to WikiKG90Mv2, where we do not provide any candidate entities for validation/test triples. Our initial experiments using the same set of baseline models, shows that WikiKG90Mv2 indeed provides a much harder challenge; our best model COMPLEX-CONCAT only achieves 0.1833 MRR on WikiKG90Mv2 as opposed to achieving 0.8637 MRR on WikiKG90M, leaving significant room for further improvement.

**Updates on `PCQM4M`.** From the KDD Cup results, we saw that the winners effectively utilized (self-calculated) 3D structure of molecules. Modeling molecular graphs in 3D space is of great interest to the graph ML community; We therefore have updated PCQM4M to PCQM4Mv2, where we provide DFT-calculated 3D structure for training molecules. For validation and test molecules, 3D structures is *not* be provided, and ML models still need to make prediction based on the 2D molecular graphs. In updating to PCQM4Mv2, we are also fixing subtle but important mismatch between some of the 2D molecular graphs and the corresponding 3D molecular graphs. Our preliminary experiments on PCQM4Mv2 suggest that the all the baseline models' MAE is improved by $\approx 0.04$ [eV] compared to PCQM4M, although the trends in model performance stay almost the same as PCQM4M.

**Updates on leaderboards.** We are introducing public leaderboards to facilitate further research advances after our KDD Cup. The test submissions of the KDD Cup 2021 were evaluated on the entire hidden test set. After the KDD Cup, we are randomly splitting the test set into two: "test-dev" and "test-challenge". The test-dev set is be used for public leaderboards that evaluate test submissions any time during a year. The test-challenge set is be left for future competitions, which we plan to hold annually to facilitate community engagement. The leaderboards have been released together with the updated datasets.

## 4 Conclusions

Modern applications of graph ML involve large-scale graph data with billions of edges or millions of graphs. ML advances on large graph data have been limited due to the lack of a suitable benchmark. Here we present OGB-LSC, with the goal of advancing state-of-the-art in large-scale graph ML. OGB-LSC provides the three large-scale realistic benchmark datasets, covering the core graph ML tasks of node classification, link prediction, and graph regression. We perform dedicated baseline analysis, scaling up advanced graph models to large graphs. We show that advanced and expressive models can significantly outperform simpler baseline models, suggesting opportunities for further dedicated effort to yield even better performance.

We used our datasets for the recent ACM KDD Cup 2021, where we have attracted huge engagement from the community and have already witnessed significant performance improvement. We summarize the winning solutions for each dataset, highliting the current best practices in large-scale graph ML. Finally, we describe how we have updated our datasets after the KDD Cup to further facilitate research advances. Overall, we hope OGB-LSC encourages dedicated community efforts to tackle the important but challenging problem of large-scale graph ML.

## Acknowledgement

We thank Michele Catasta and Larry Zitnick for helpful discussion, Shigeru Maya for motivating the project, Adrijan Bradaschia for setting up the server for the project, and Amit Bleiweiss, Benjamin Braun and Hanjun Dai for providing helpful feedback on our baseline code, and the DGL Team for hosting our large datasets.

Stanford University is supported by DARPA under Nos. N660011924033 (MCS); ARO under Nos. W911NF-16-1-0342 (MURI), W911NF-16-1-0171 (DURIP); NSF under Nos. OAC-1835598 (CINES), OAC-1934578 (HDR), CCF-1918940 (Expeditions), IIS-2030477 (RAPID); Stanford Data Science Initiative, Wu Tsai Neurosciences Institute, Chan Zuckerberg Biohub, Amazon, JPMorgan Chase, Docomo, Hitachi, JD.com, KDDI, NVIDIA, Dell, Toshiba, Intel, and UnitedHealth Group. Weihua Hu is supported by Funai Overseas Scholarship and Masason Foundation Fellowship. Matthias Fey is supported by the German Research Association (DFG) within the Collaborative Research Center SFB 876 "Providing Information by Resource-Constrained Analysis", project A6. Hongyu Ren is supported by Masason Foundation Fellowship and Apple PhD Fellowship. Jure Leskovec is a Chan Zuckerberg Biohub investigator.

Our baseline code and Python package are built on top of excellent open-source software, including NUMPY (Harris *et al.*, 2020), PYTORCH (Paszke *et al.*, 2017), PYTORCH GEOMETRIC (Fey and Lenssen, 2019), DGL (Wang *et al.*, 2019), and DGL-KE (Zheng *et al.*, 2020).

The HOKUSAI facility was used to perform some of the quantum calculations. This work was supported by the Japan Society for the Promotion of Science (JSPS KAKENHI Grant no. 18H03206). We are also grateful to Maeda Toshiyuki for helpful discussions.

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
