# OpenReview forum: "OGB-LSC: A Large-Scale Challenge for Machine Learning on Graphs"
_NeurIPS.cc/2021/Track/Datasets_and_Benchmarks/Round2 — NeurIPS 2021 Datasets and Benchmarks Track (Round 2)_

### Official Review · Reviewer_Kmm6 · 2021-09-16
**Solid and impactful contribution to graph learning community**

**Rating:** 8
**Confidence:** 4
**Correctness:** No technical flaws are spotted.
**Clarity:** Yes.

**Strengths:**

+ Large-scale evaluation of graph neural networks is a very important direction. The authors collect three datasets from various domains at different levels, which will facilitate development of algorithms for both academia and industry.

+ The collected datasets have been verified through a KDDCup conference that have attracted a lot of competitors around world, which demonstrates its real-world impact.

+ Detailed discussions of baseline performance, common practices, and winning solutions are provided.

**Weaknesses:**

I am overall satisfactory with the presentation by the authors. Please see my minor comments in the "additional question" section.


**Additional Feedback:**

- The authors may discuss how the data is cleaned and how to ensure the label quality, especially for the MAG240M dataset, where arXiv subject areas may not be always accurate and moderators often change the initial category chosen by the authors.

- Since there are different challenges for different datasets, I wonder whether the OGB team will adopt more evaluation metrics, not only limiting to performance. For example, I am in particular interested in resource-constrained scenarios for giant graphs, considering common practices like the neighborhood sampling are computation-intensive.

- Some key takeaways summarizing common practices could be provided at the end of Section 2.


**Documentation:**

Yes. The documentation is comprehensive.

**Relation To Prior Work:**

Large-scale graph datasets are lacking in current open literature.

**Summary And Contributions:**

This paper presents the OGB-LSC datasets, a large-scale dataset collection that focuses on node-, link-, and graph-level predictions. Baseline performance and interesting findings regarding experimental results have been provided. Future update plans are also discussed with preliminary observations.

---

> ### Author Response · Authors · 2021-09-29
> **Reply**
>
> We thank the reviewer for the positive feedback. Below are our responses to the reviewer’s suggestions/questions.
>
> **1. Correctness of the arXiv labels**
>
> We use the category reviewed by the moderator, not the category assigned by the authors.
>
> **2. Evaluation metrics beyond the model accuracy.**
>
> Including system-oriented metrics is definitely of interest. Especially, the metric in the spirit of the DAWNBench [1] (i.e., the time needed to achieve the threshold performance given a fixed model and training strategy) would be interesting to include. We are planning to investigate this direction once the community has more or less come to a consensus on the model architecture and training strategy on the large graphs.
>
> [1] Coleman, Cody, et al. "Dawnbench: An end-to-end deep learning benchmark and competition." Training 100.101 (2017): 102.
>
> **3. Summarize key takeaways in Section 2**
>
> We are not sure if we understand the reviewer’s suggestion. We have already summarized the per-dataset key takeaways at the end of each dataset subsection in Section 2. As the datasets are drastically different from each other, it would be hard to draw shared takeaways across all the datasets.

---

### Official Review · Reviewer_S925 · 2021-09-20
**Datasets and Benchmarks for Learning on Large Graphs**

**Rating:** 8
**Confidence:** 4
**Correctness:** I have not found any issues with corr…
**Clarity:** The paper is clear and well-written.

**Strengths:**

1. MAG240M, WikiKG90M, PCQM4M are actually large — this is, in my opinion, the main and solid contribution of this paper.
2. Each dataset has reasonable and realistic splits.
3. Carefully conducted baseline evaluation. The authors describe the hyper-parameter selection process and evaluate each method with different depth, hidden dimensionality, and other parameters. In addition, the authors include simple baselines such as graph-agnostic MLP and label propagation. Including these baselines allows evaluating the impact of using graph deep learning methods. Also, this evaluation might reveal peculiarities of methods for learning on large graphs and compares all the state-of-the-art approaches under similar realistic conditions. This contribution is crucially important for the graph ML community.

**Weaknesses:**

Although I think that the following do not downplay the paper's main contributions, I find it necessary to point them out to start a reasonable discussion.
1. The paper suffers from the lack of the related work review and comparison with other datasets. The authors state that the introduced datasets are a magnitude larger than the existing ones but do not provide a corresponding comparison.
2. The paper nor the supplementary materials do not provide the graphs' statistics such as graph diameter, connectivity, etc.
3. The provided datasets look like the (significantly) larger of the existing ones — citation networks, Wikipedia articles, that are significantly different from the industrial data, e.g., transaction networks. This issue does not downplay the impact of this benchmark but potentially could raise a concern about its applicability to the industrial conditions.
4. The nodes' features of MAG240M and WikiKG90M are obtained through a sentence encoder. This reveals several concerns: first, as was mentioned above, the real-world datasets often contain heterogeneous features such as categorical data; second, the benchmark is encoder dependent, but, as far as I know, the study on encoder choice was not conducted as well as fine-tuning the encoder along with the training baseline models. Also, it would be interesting to see pure text classification baselines here, like fine-tuning SOTA transformers.
5. Although the authors show the plans on publishing the 3D structure information for PCQM4M, this data is essential for learning on molecular graphs and is not presented in the current version. Many methods that incorporate the graph's 3D structure or angular information were developed (some examples are [1, 2, 3]) and showed their superiority over straightforward graph DL approaches and found applications in real-world tasks such as CASP MQA. So, it would be interesting to see a comparison of these methods on large-scale data, but it seems to be hard without having 3D structures for the test split.

[1] SchNet: a continuous-filter convolutional neural network for modeling quantum interactions. K. T. Schütt et al. NIPS’17

[2] Directional Message Passing for Molecular Graphs. J. Klicpera et al. ICLR 2020

[3] Spherical convolutions on molecular graphs for protein model quality assessment. I. Igashov et al. Machine Learning: Science and Technology, Vol. 2, Num. 4


**Additional Feedback:**

No additional feedback.

**Documentation:**

A submission contains all the necessary documentation.

**Ethics:**

I don't have any ethics concerns about this paper.

**Relation To Prior Work:**

The relation to prior work is clearly discussed.

**Summary And Contributions:**

This paper introduces three large-scale graph datasets that are significantly larger than other existing datasets. Each dataset corresponds to different graph ML tasks: node level, link level, and graph level. Exploring how graph-based machine learning methods operate on large-scale data is essential for the field. However, there are few open benchmark datasets of the size comparable to the industrial data. This paper fills this empty room and, in addition to datasets themselves, brings a benchmark of the state-of-the-art methods of learning on large graphs.

---

> ### Author Response · Authors · 2021-09-29
> **Reply (2/2)**
>
> **5. Comparison against joint training of GNN and the text encoder**
>
> Joint training of the text encoder and GNN would most likely provide performance gain but could be prohibitively computationally expensive in practice. To give the reviewer some sense, to generate just a single node embedding, around 250 neighboring node embeddings need to be aggregated, which would require RoBERTa to encode the corresponding 250 titles/abstracts. For a 45GB high-end GPU, the maximal batch size it can fit for RoBERTa is 48, so we will need 5 high-end GPUs just to generate a single node embedding in a backpropagatable manner, which is way too expensive in practice.
>
> **6 Comparison with fine-tuned RoBERTa**
>
> Thank you for the insightful suggestion. It is indeed interesting to see the direct fine-tuning of RoBERTa over the labeled arXiv papers (without utilizing the graph structure).
>
> We fine-tuned the RoBERTa-base model and got the validation accuracy of 66.89% (trained for 2.5 days on a single 45GB GPU). The accuracy is indeed much better than our MLP+non-finetuned-RoBERTa baseline (52.67% validation accuracy, 5 minutes of training on a single 11GB GPU), but fine-tuning RoBERTa is also significantly more expensive.
> Compared to graph models, fine-tuned RoBERTa gives comparable performance to our GraphSAGE baseline (66.79 % validation accuracy, 8 hours of training on a single 11GB GPU) and is much worse than the SoTA graph model (73.7% single model validation accuracy by the BD-PGL Team). Overall, these results suggest that fine-tuned text encoder does give a strong performance, but graph-based models are even stronger without fine-tuning the RoBERTa encoder (which is computationally expensive).
>
> **7. 3D structure data for PCQM4Mv2**
>
> As we noted in the general response, we have made everything publicly available, including the requested 3D molecule data (link: http://ogb-data.stanford.edu/data/lsc/pcqm4m-v2_xyz.zip, 2.7GB).
>
> As pointed out by the reviewer, there are indeed many promising GNNs developed for 3D molecules. However, as the reviewer is already aware, these GNNs require 3D structure to be available at inference time, which is often hard to obtain in practice; hence, we assume 3D structure is not available at inference time. It is largely an open question how to utilize 3D structure of training molecules to improve the performance on the final prediction task. One potential idea could be to learn a model to generate 3D structure and then use the state-of-the-art 3D GNN to make the final prediction. We hope the community will come up with innovative solutions to this practical and novel setting.

---

> ### Author Response · Authors · 2021-09-29
> **Reply (1/2)**
>
> We thank the reviewer for the positive and constructive feedback. Below, we address each discussion point raised by the reviewer. We have also updated our paper accordingly, which further improves our paper.
>
> **1. Comparison with related datasets.**
>
> Below we compare the OGB-LSC datasets with the comparable original OGB datasets (that have been considered “large” so far). We have added these objective comparisons in footnote 1 of the paper.
>
> Node-level classification on a heterogeneous graph
>
> - MAG240M #nodes: 244,160,499
> - ogbn-mag #nodes: 1,939,743
> - MAG240M is 126 times larger than ogbn-mag in terms of #nodes
>
> KG completion on WikiData
>
> - WikiKG90M #nodes: 87,143,637
> - ogbl-wikikg2 #nodes: 2,500,604
> - WikiKG90M is 35 times larger than ogbl-wikikg2 in terms of #nodes
>
> Molecular property prediction
>
> - PCQM4M #graphs:  3,803,453
> - ogbg-molpcba #graphs: 437,929
> - PCQM4M is 9 times larger than ogbg-molpcba in terms of #graphs
>
> **2. Basic graph statistics**
>
> As suggested by the reviewer, we have computed the basic statistics, such as clustering coefficients and average node degrees, using the SNAP library. As pre-processing, we converted all the graphs into undirected homogeneous graphs. We cannot report graph diameter as it is too expensive to compute on our large graphs. Some statistics are still being computed, and we will update them by the camera-ready of the paper. We have added Table 8 in Appendix B to include the basic graph statistics.
>
> **MAG240M**
> - Avg degree: 14.15
> - Avg clustering coefficient: 0.033
>
> **MAG240M (homogenized)**
> - Avg degree: 21.30
> - Avg clustering coefficient: 0.031
>
> **WikiKG90M**
> - Avg degree: 10.93
> - Avg clustering coefficient: Still computing.
>
> **WikiKG90Mv2**
> - Avg degree: 12.59
> - Avg clustering coefficient: Still computing.
>
> **PCQM4M**
> - Avg num nodes:  14.15
> - Avg num edges:  14.57
> - Avg diameter:  7.96
> - Avg clustering coefficient:  0.010
> - Avg node degrees:  2.05
> - Avg maximum SCC size:  1.0
>
> **PCQM4Mv2**
> - Avg num nodes:  14.14
> - Avg num edges:  14.56
> - Avg diameter:  7.95
> - Avg clustering coefficient:  0.011
> - Avg node degrees:  2.05
> - Avg maximum SCC size:  1.0
>
> **3. Difference to the real-world industrial data**
>
> As we have already mentioned in the “limitation” paragraph of Appendix A, many industrial large graphs, including transaction networks and social networks, cannot be made publicly available due to privacy and cooperative concerns, making it hard to include them in our benchmark. That being said, it is our hope that many methodological insights on our large graphs (training strategy, GNN architecture, regularization, etc) still transfer well to those graphs. We leave the thorough investigation to future work. In Appendix A, we have further expanded our discussion about this limitation.
>
>
> **4. Dependency on the Text encoder**
>
> As the reviewer pointed out, graph models can indeed depend on the choice of the text encoder. We provide the pre-encoded text features in order to
>
> (1) Simplify the setting
>
> (2) Let the community focus on the graph aspect of the problem.
>
> (3) Keep the integrity of the hidden test labels; see the rules here.
>
> Moreover, RoBERTa is considered a strong sentence encoder and is representative of the current best practice of encoding texts into vectors. Advances in language modeling would probably provide a better language encoder but it is our belief that language encoder and graph encoder are somewhat complementary. The heterogeneous features are beyond the scope of our benchmark, and we leave it for future work.

---

### Official Review · Reviewer_4P8S · 2021-09-22
**A good set of realistic, large ML graph benchmarks, with datasets, baselines, and high performance results**

**Rating:** 9
**Confidence:** 4
**Correctness:** The datasets, benchmarks, baselines a…
**Clarity:** The work is very clearly presented.

**Strengths:**

- The main strengths of the work are:
    - Providing three large, realistic datasets for ML graph tasks
    - Providing results of multiple competitive baselines on those tasks
    - Providing documentation and code to run those baselines
    - Providing results of state-of-the-art winning entries of Kaggle challenges on those same results
    - A clear, well written paper

**Weaknesses:**

- I couldn't find any relevant weakness.
- There is a typo on line 98 (annotating → annotate)

**Additional Feedback:**

Great work, congratulations!

**Documentation:**

The work is well documented, with supplementary material and a Github site including code, data, and installation and running instructions.

**Ethics:**

No ethics concerns.

**Relation To Prior Work:**

The relation with prior work is clear.

**Summary And Contributions:**

The Open Graph Benchmark - Large Scale Challenge (OGB-LSC) is a set of three large real-world datasets (between 55M and 1.7B edges) focusing on three different graph ML task types (node-, link-, and graph-level), and including the task metrics, competitive baselines, and state-of-the-art reference results (from Kaggle challenges winning entries). The datasets, baseline code, and documentation is also made available. The OGB-LSC fills a gap in the graph ML community by providing datasets and competitive results at a very large scale.

---

> ### Author Response · Authors · 2021-09-29
> **Reply**
>
> We thank the reviewer for the positive feedback! The typo has been fixed in the updated version.

---

### Author Response · Authors · 2021-09-29
**General response. Datasets updated and Leaderboards released.**

We thank the reviewers for the positive and constructive feedback. We are glad that all the reviewers find our benchmark important in advancing large-scale graph ML. We reply to each reviewer to address their main concerns/suggestions. We have also updated our paper accordingly, which we hope to further improve our paper.

In addition, as promised in the submitted version of our paper, we have released the new package version (1.3.2 available at the master branch at https://github.com/snap-stanford/ogb) and public leaderboard infrastructure (https://ogb.stanford.edu/docs/lsc/leaderboards/).
This means that the updated datasets (WikiKG90Mv2 and PCQM4Mv2) and their baseline experiments are now publicly available for the community to explore. Appendix B has been also updated, explaining the updates of our datasets and baseline experiments.

---

### Decision · Program_Chairs · 2021-10-09

**Decision:**

Accept

**Comment:**

In this paper the authors present three real-world datasets in different domains to test graph ML capabilities in three distinct tasks. The authors present baselines and learnings from these datasets being used in the 2021 KDD Cup. All reviewers found the paper to be strong and valuable, and as such I believe it should be accepted.